# Predictors of return to work among women with long-term neck/shoulder and/or back pain: A 1-year prospective study

Mamunur Rashid[1]*, Marja-Leena Kristofferzon[2], Annika Nilsson[2]

1 Department of Public Health and Sports Sciences, Faculty of Health and Occupational Studies, University of Gävle, Gävle, Sweden, 2 Department of Caring Sciences, Faculty of Health and Occupational Studies, University of Gävle, Gävle, Sweden

* mamunur.rashid@hig.se

## Abstract

### Background

Sick leave due to musculoskeletal pain, particularly in the neck/shoulders and back, is one of the major public health problems in Western countries such as Sweden. The aim of this study was to identify predictors of return to work (RTW) among women on sick leave due to long-term neck/shoulder and/or back pain.

### Methods

This was a prospective cohort study with a 1-year follow-up. The study participants were recruited from a local Swedish Social Insurance Agency register and had all been on sick leave for $\geq$ 1 month due to long-term ($\geq$ 3 months) neck/shoulder and/or back pain. Data on predictors and outcome were collected using a self-administered questionnaire. A total of 208 women aged 23–64 years were included at baseline, and 141 responded at the 1-year follow-up. Cluster analyses were performed to identify one predictor from each cluster for use in the regression model.

### Results

At the 1-year follow-up, 94 of the 141 women had RTW and 47 had not. Women who engaged in more coping through increasing behavioral activities (OR: 1.14, 95% CI: 1.03–1.25) and those who more strongly believed they would return to the same work within 6 months (OR: 1.22, 95% CI: 1.10–1.37) had an increased probability of RTW. Receiving more social support outside work (OR: 0.50, 95% CI: 0.28–0.92) decreased the odds of RTW at the 1-year follow-up.

### Conclusions

Behavioral activities, beliefs about returning to the same work, and social support outside work were predictors of RTW at the 1-year follow-up. Healthcare professionals should

**Data Availability Statement:** In Sweden, there is an ethical restriction on sharing datasets due to sensitive patient information. The information about data confidentiality was mentioned in the

information letter and that was sent to the participants for obtaining their written consent. The ethics approval from the ethics committee and information letter has been attached as supporting information (S2A and S2B) (in Swedish). Moreover, the contact information of the ethics committee as follows: registrator@uppsala.epn.se.

**Funding:** The project was supported by the University of Gävle, Sweden. The funders had no role in study design, data collection and analysis, decision to publish, or preparation of the manuscript.

**Competing interests:** The authors have declared that no competing interests exist.

consider these predictors in their efforts to prevent prolonged sick leave and to promote RTW in this population.

## Introduction

Work is an important part of life and has a fundamental role in physical health and psychosocial well-being [1]. Working life may be interrupted by musculoskeletal pain (MSP) that can lead workers to take sick leave [2]. MSP occurs predominantly in the neck/shoulders and back and more frequently in women than in men, and has been considered the second most frequent reason for sick leave in Western countries [2–6]. According to a report from the Swedish Social Insurance Agency, 61% of the Swedish population on sick leave for MSP are women [7]. Individuals on sick leave for MSP may experience positive or negative consequences of sick leave [8], depending on how they relate to their sick role during sick leave periods [8, 9]. The experiences of women on long-term sick leave may differ; some experience hopelessness and lack of motivation, whereas others may make plans for future work and seek support [10]. Being on long-term sick leave per se has been considered a predictor of future poor physical health, low psychosocial well-being, and reduced work ability [11]. Further, sick leave results in medical expenses, workers' compensation, and productivity loss in Western countries, including Sweden [12, 13]. Considering the enormous economic cost, there is a need to examine factors that may obstruct or promote return to work (RTW) among women on sick leave for long-term MSP.

Ways of assessing RTW include work status, time until the suspension of time-loss benefits, and time until claim closure [14, 15]. The measures used vary with the population studied as well as across societies and countries, which have different healthcare systems and workers' compensation regulations [14–16]. In the present study, RTW was assessed in terms of work status; that is, whether individuals were working or not, and the extent of the work. RTW following MSP is a multifaceted process that is not merely dependent on physical health and/or ability; rather, the process of RTW in this regard involves an individual's resources, such as recovery beliefs and coping strategies. These resources are related to gradually increasing one's work ability in order to be able to cope with work demands and life events such as stress and pain [17].

Several potential factors related to RTW among individuals with MSP have been studied previously. For example, psychological factors such as recovery beliefs, anxiety, depression, locus of control, and health-related quality of life have been shown to be associated with RTW among people with MSP [18–21]. Studies have also shown that RTW may be affected by pain-related behaviors such as pain intensity and fear-avoidance beliefs, and work-related factors such as job stress and job satisfaction [21, 22]. One study found that age, gender, motivation, coping, and general health status were important for RTW on the individual level [23]. Earlier research has also shown that individuals' belief in their ability to work in the future is a predictor of RTW [24, 25], and that increased work ability per se is an important predictor for RTW among women with pain in the neck/shoulder and/or back [26]. Furthermore, social support from work and outside work influences recovery from long-term MSP [27], which may have a significant role in RTW [28]. Several studies have reported that social support outside work, such as close relationships and supportive social environment, is associated with reduced MSP. This may increase work ability, which could, in turn, contribute to RTW [26, 29]. Social support either from work or outside work was also found to be a positive indicator in the process of RTW [30, 31]. Finally, one study found that social support from a partner relationship might not be related to RTW among female workers on sick leave [32]. It seems that existing

previous studies regarding social support and RTW have given mixed results, depending on the study population and measurement of social support.

Most of the previous studies on predictors of RTW among individuals with MSP have been conducted on workers in general, or only on male workers, and have focused on acute and sub-acute pain or non-specific low back pain [19–24, 28]. A previous systematic review [33] aimed at summarizing prognostic factors related to RTW among people with long-term neck/ shoulder or back pain suggested that perceived health, recovery beliefs and work ability may be predictive of RTW. However, the authors concluded that their findings were based on only a few studies, and none of the studies in the review focused solely on women [33]. Factors associated with RTW may vary between men and women even if they are in the same line of work [34]. Moreover, women still have the highest proportion of sick leave for long-term MSP [7]. Because being on sick leave for long-term MSP is a problem from an individual, social and financial perspective, it is important to identify predictors of RTW that could be considered in rehabilitation for this population [35]. Thus, the aim of the present study was to identify predictors of RTW among women on sick leave for long-term neck/shoulder and/or back pain.

## Methods

### Study design and settings

This was a prospective cohort study with a 1-year follow-up. The study participants were sampled from a local Swedish Social Insurance Agency register which covered all people receiving sick leave benefits in Central and Northern Sweden. The periods of data collection were the spring of 2016 (baseline) and the spring of 2017 (follow-up).

### Study sample

The participants were selected by the Swedish Social Insurance Agency based on their medical certificate, which had been issued by their primary healthcare or hospital physician. Before the selection procedure, two of the authors (MLK and AN) instructed personnel at the Swedish Social Insurance Agency on how to select participants. To qualify for inclusion in the study, a participant had to be a woman aged 18–65 years, on $\geq$ 50% sick leave from her usual employment (i.e., she could be working part-time), and on sick leave for $\geq$ 1 month due to long-term neck/shoulder and/or back pain that had lasted for $\geq$ 3 months. Neck/shoulder and/or back pain were classified to the following diagnostic codes from version 10 of the International Classification of Diseases: M53.1 (cervicobrachial syndrome), M54.2 (cervicalgia), M54.4 (lumbago with sciatica), M54.5 (low back pain), M54.9 (dorsalgia unspecified), M75.8 (other shoulder lesions), M75.9 (shoulder lesion, unspecified), and M79.1 (myalgia). The code for myalgia (M79.1) was included because myalgia pain can spread to neck, shoulders, and back; for example, trapezius myalgia is characterized by acute or persistent neck/shoulder pain. Because the specific cause of MSP is often uncertain, many diagnostic codes are used for this population. The diagnostic codes were selected based on a previous study [36] and discussions with the Swedish Social Insurance Agency. Understanding the Swedish language was also required for the participants to complete the questionnaire. Women were excluded from the study if they had been diagnosed with rheumatoid arthritis, multiple sclerosis, stroke, cancer, Parkinson's disease, bipolar disease, or schizophrenia, or were pregnant. These diseases, disorders, and conditions were chosen as exclusion criteria because individuals affected by them require different types of interventions and hence may have a different RTW process [37]. In addition, women who had early retirement (i.e., before 65 years of age) were not included in the study. The project was approved by the Regional Ethical Review Board in Uppsala, Sweden (Reg. no. 2.3.2-2015/548).

## Data collection

An initial invitation letter and a self-administered questionnaire including eight instruments were sent to the participants by the Swedish Social Insurance Agency. Several demographic variables such as age, cohabitation, number of children, education, years in the workforce, type of work, stress in the last 6 months, life-long pain duration, and physical activity were included in the questionnaire. In addition, a pain figure was included to collect information on the location of pain on the body [38]. Participants who agreed to take part in the study returned the questionnaire along with a signed informed consent form. To increase the response rate, two reminders were sent about two weeks apart. At the 1-year follow-up, the same questionnaire was sent to the baseline participants, this time including two additional background questions meant to detect RTW status.

## Candidate predictors

Based on a systematic review [33] and previous empirical research [19, 20, 39] on factors associated with RTW among people with MSP, the candidate predictors of RTW presented below were considered in this study.

The Coping Strategies Questionnaire [40] consists of eight subscales. In the present study, two subscales, "increase behavioral activity" and "ignore sensations", were used to assess: (i) coping through increasing behavioral activities such as leisure activities, reading, and socialization; and (ii) coping by ignoring sensations, for example by relaxing, thinking pleasant thoughts, and praying [40]. These two subscales were chosen because they are commonly used in MSP patients [41, 42] and represent both cognitive and behavioral coping processes. The sub-scale "pain catastrophizing" was not selected for the present study because we did not measure RTW in terms of any direct pain-related outcome such as reduction in pain and disability [43]. Each scale was measured using six items rated on a 7-point Likert scale (0 = never; 6 = always). An index of each scale was obtained by calculating the sum of all scores of the six items, such that higher values represented more frequent use of the coping strategies. Cronbach's α was 0.86 for each subscale [44].

Self-efficacy was measured using the General Self-Efficacy scale [45], which consists of 10 items rated on a 4-point Likert scale (1 = not true; 4 = completely true). Total scores on the GSE range from 10 to 40 points, with higher values representing greater self-efficacy. Cronbach's α for this scale was 0.92 [44].

Sense of coherence was measured using a short version of the Sense of Coherence scale [46], which consists of 13 items rated on a 7-point scale (1 = never; 7 = very often). The total scores range from 13 to 91 points, with higher scores indicating greater sense of coherence. Cronbach's α for this scale was 0.84 [44].

Physical activity was assessed using a single question: "How often do you exercise regularly for at least 30 minutes, e.g., walking, jogging, swimming, cycling, or walking in the garden?" Respondents answered the question by selecting one of four alternatives: 0 days/week, 1–3 days/week, 4–5 days/week or 6–7 days/week.

Beliefs about returning to the same work within 6 months were also assessed using a single question: "Do you believe you will return to the same work within 6 months?" Respondents answered on a 10-point Likert scale (1 = highly unlikely to return to the same work; 10 = highly likely to return to the same work).

Pain intensity was measured using three items from the Multidimensional Pain Inventory [47]: (i) How much pain are you experiencing right now? (ii) How much pain have you experienced on average during the past week? (iii) How much do you suffer from your pain? The participants rated each item on a 7-point Likert scale (0 = no pain; 6 = extreme pain). An

index was created by calculating an average value of the items, with higher values indicating higher pain intensity. Cronbach's α for this scale was 0.76 [44].

Social support outside work was measured using the social support subscale from the Multi-dimensional Pain Inventory, which consists of three items [47]: (i) When you are experiencing pain, how much support or help do you get from the people closest to you (family or friends)?, (ii) How concerned are your friends and family about the pain you experience?, (iii) How much consideration do your friends and family members show for your pain? Respondents rated the items on a 7-point scale (0 = not at all; 6 = very much). An index was created by calculating an average value of the items, with higher values indicating higher social support. Cronbach's α for this scale was 0.60.

Depression was assessed using the Hospital Anxiety and Depression Scale [48], which consists of 14 items rated on a 4-point Likert scale; all of the even-numbered items form a depression subscale. Total scores on this subscale range from 0 to 21 points, with higher values indicating greater depression. Although the scale has two subscales (anxiety and depression), depression was chosen for this study because depression is more common in women than in men [49]. In addition, depression was found to be negatively correlated with well-being in the same population in a previous study, but this was not the case for anxiety [44]. Cronbach's α for depression subscale was 0.91 [44].

The Demand Control Support Questionnaire [50] was used to measure job strain. This consists of four subscales: psychosocial demands (5 items), skills discretion (2 items), decision authority (4 items), and social support (6 items). For each item, responses were made on a 4-point Likert scale ranging from 1 (strongly agree) to 4 (strongly disagree). To capture job strain, an index was constructed for each of the psychological demands, skills discretion, and decision authority subscale. Skills discretion and decision authority were then merged into one scale called decision latitude. Afterwards, a job strain score was created by calculating the ratio between psychological demands and decision latitude, with higher values representing higher job strain [51]. Cronbach's α for this was 0.57 [44].

Life-long pain duration was assessed using the single question: "How long have you been experiencing pain?" The response was reported in months.

## Outcome measure

Information on RTW was gathered at the 1-year follow-up using two questions: "Are you working right now?" and "To what extent are you working?". RTW status was considered a dichotomous variable. If participants worked > 50% of their extent of employment at baseline, they were categorized as RTW; otherwise, they were categorized as not RTW (NRTW).

## Statistical analysis

Descriptive statistics for the demographic variables are presented as proportions, means, and standard deviations. Difference in means of RTW and NRTW by baseline characteristics and study variables were evaluated with a chi-squared test for binary variables, Fisher's exact test for nominal and ordinal variables, and an independent t-test for numerical variables. Scatter-plots showed that all variables were approximately normally distributed, and there were no outliers in the data. An attrition analysis was performed between participants and dropouts at the 1-year follow-up considering age, life-long pain duration, pain intensity, work ability, and well-being.

Because the number of participants limited the number of predictors that could be included in the prediction model [52], we had to select a subset of predictors among the candidates. To

reduce the risk of modeling spurious relationships, the selection of predictors was made without empirically verifying their relationship with the outcome RTW [53, 54].

To reduce the number of predictors, the following steps were carried out. First, a hierarchical cluster analysis was performed on all candidate predictors using squared Euclidean distance and average linkage between clusters [53–55]. Second, one predictor from each cluster was chosen on the basis of (i) a formal statistical test which showed the relative dispersion of its values in the within-cluster sample, and (ii) the importance of previous findings in relation to RTW in this regard. Finally, multiple logistic regression was performed to estimate the association between the selected predictors and RTW. Age was controlled for in the adjusted analysis. Nagelkerke's $R^2$ was used to measure the overall predictive ability of the selected predictors of RTW in the logistic regression. The level of significance was set at $p < 0.05$. All data analyses were performed using version 24 of the IBM SPSS statistical software package.

## Results

Of the 600 women initially contacted, 275 responded to the survey. Sixty-seven were excluded based on the exclusion criteria, resulting in 208 women at baseline. After 1 year, a follow-up survey was sent to the baseline participants and 141/208 women responded, corresponding to a response rate of 68% (Fig 1). An attrition analysis on age, life-long pain duration, pain intensity, work ability, and well-being showed no significant differences in baseline values between the dropouts and those remaining at follow-up (S1 Table). There was no multi-collinearity between the predictors in the prediction model, as the variance inflation factor was ≤ 1.04 [56].

Table 1 represents the baseline characteristics of the 94 participants who had RTW and the 47 who had NRTW at the 1-year follow-up. The group who had RTW rated less pain intensity, more coping through increasing behavioral activities and more strongly believed in returning to work within 6 months. Conversely, the group who NRTW received more social support outside work. There were also significant differences between the groups concerning age, cohabitation, and economic situation.

Fig 2 (based on the dendrogram in S1 Fig) shows the cluster analysis. Although four predictors could fit into the prediction model, we chose three (plus age as a covariate) because the difference in distance (i.e., squared Euclidean distance of z-scores) between clusters in the 3-cluster and the 4-cluster solution was small. Cluster I consisted of coping through increasing behavioral activities, ignore sensations, self-efficacy, and a sense of coherence. Increasing behavioral activities was chosen from the cluster because it had the highest relative dispersion in values. Moreover, the results of a previous study support this choice, because using behavioral activities as a coping strategy might help women deal with their pain [57]. Cluster II consisted of physical activity and beliefs about returning to the same work within 6 months. Both variables had the same dispersion in values, and so belief about returning to the same work within 6 months was selected on the basis of previous research showing that such beliefs are important for RTW [23]. Furthermore, earlier studies in this population found that beliefs about returning to the same work within 6 months was correlated with work ability [44], and that work ability predicted RTW [58]. Cluster III consisted of pain intensity, social support outside work, depression, job strain, and pain duration. Social support outside work was chosen from this cluster because it had the highest relative dispersion in values. According to previous studies, women on sick leave for MSP may need social support from immediate family, relatives and the surrounding environment to cope with pain and to assist in planning for RTW [30].

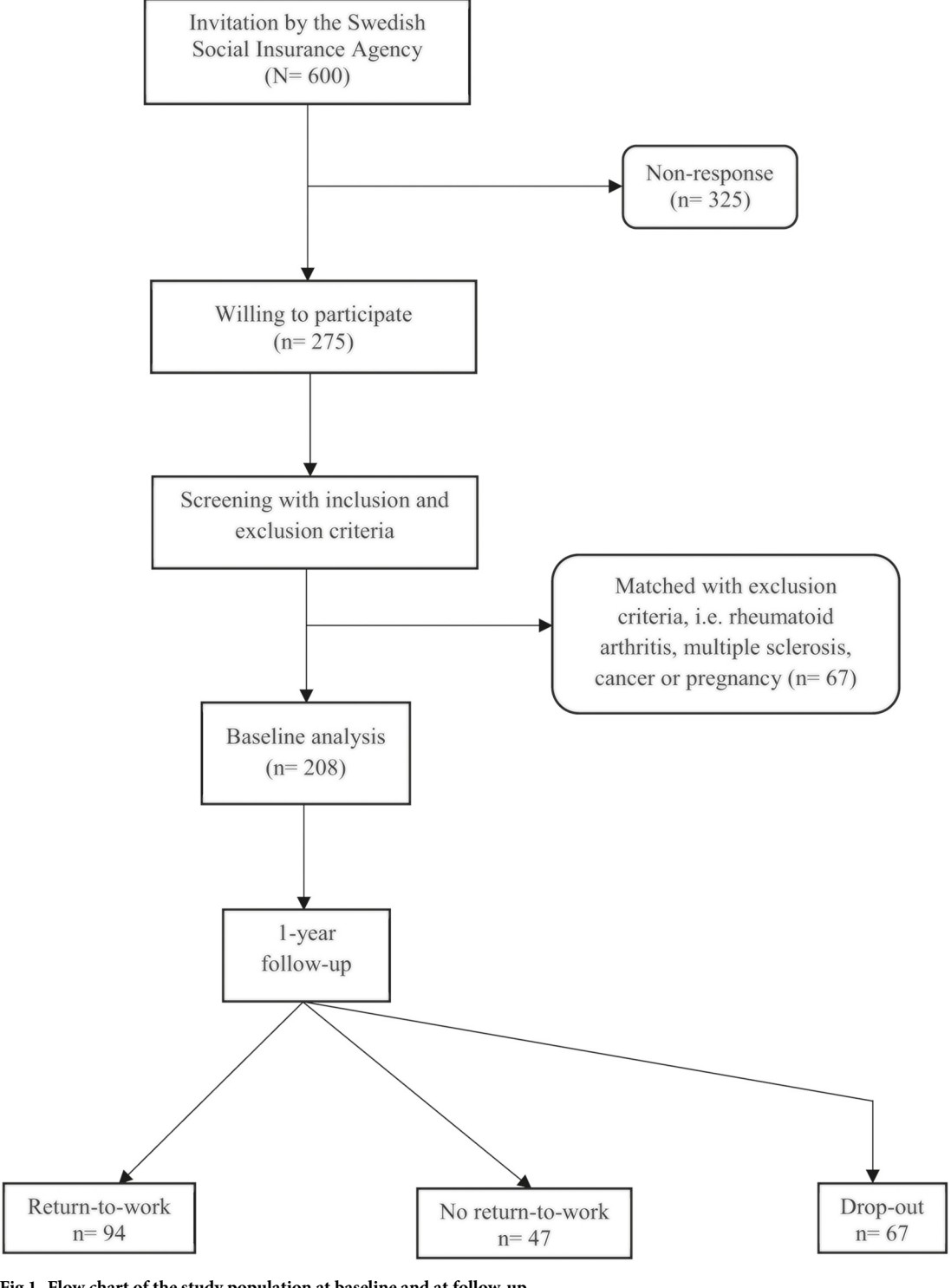

**Fig 1. Flow chart of the study population at baseline and at follow-up.**

**Table 1. Baseline characteristics and study variables of participants who had returned to work (RTW) and who had not returned to work (NRTW) at the 1-year follow-up.**

| Baseline characteristics | RTW (n = 94) | NRTW (n = 47) | *p*-value |
|---|---|---|---|
| Age (M, range), years | 49.04 (23–63) | 53.51 (24–64) | **0.007** |
| Cohabitation, n (%) | | | **0.04** |
| Living with partner | 75 (79.8) | 30 (63.8) | |
| Living alone | 17 (18.1) | 12 (25.6) | |
| Living apart | 2 (2.1) | 5 (10.6) | |
| Children living at home, n (%) | | | 0.37 |
| No | 53 (56.4) | 29 (64.4) | |
| Yes | 41 (43.6) | 16 (35.6) | |
| Education, n (%) | | | 0.69 |
| Elementary | 13 (13.8) | 10 (21.3) | |
| Upper secondary | 45 (47.9) | 22 (46.8) | |
| University | 32 (34.0) | 14 (29.8) | |
| Others | 4 (4.3) | 1 (2.1) | |
| Economic situation, n (%) | | | **0.004** |
| Very dissatisfied | 5 (5.8) | 9 (19.2) | |
| Dissatisfied | 19 (20.4) | 8 (17.0) | |
| Acceptable | 36 (38.5) | 21 (44.7) | |
| Good | 27 (28.7) | 5 (10.6) | |
| Very good | 6 (6.6) | 4 (8.5) | |
| Years in the workforce (M, range) | 30.04 (6–46) | 32 (3–47) | 0.25 |
| Type of work[1], n (%) | | | 0.71 |
| White-collar | 35 (37.2) | 16 (34.0) | |
| Blue-collar | 59 (62.8) | 31 (66.0) | |
| Stress in the last 6 months | | | **0.04** |
| All of the time | 11 (11.8) | 12 (26.7) | |
| Almost all of the time | 24 (25.8) | 16 (35.6) | |
| Some of the time | 41 (44.1) | 14 (31.1) | |
| A small part of the time | 15 (16.1) | 2 (4.4) | |
| Not at all | 2 (2.2) | 1 (2.2) | |
| Pain area, n (%) | | | 0.52 |
| Neck/shoulders | 65 (69.1) | 32 (68.1) | |
| Back | 63 (67.0) | 38 (80.9) | |
| Neck/shoulders and back | 36 (25.5) | 25 (17.7) | |
| Behavioral activity[2] (M ± SD) | 13.10 ± 4.9 | 11.13 ± 5.0 | **0.03** |
| Ignore sensations[2] (M ± SD) | 13.29 ± 5.0 | 13.69 ± 6.4 | 0.69 |
| Self-efficacy (M ± SD) | 30.62 ± 4.4 | 29.26 ± 7.0 | 0.23 |
| Sense of coherence (M ± SD) | 63.65 ± 12.5 | 58.50 ± 13.16 | **0.03** |
| Physical activity, n (%) | | | 0.25 |
| 0 days/week | 12 (12.8) | 7 (15.0) | |
| 1–3 days/week | 48 (51.1) | 16 (34.0) | |
| 4–5 days/week | 24 (25.5) | 12 (25.5) | |
| 6–7 days/week | 10 (10.6) | 12 (25.5) | |
| Beliefs about returning to work[3] (M ± SD) | 7.69 ± 3.2 | 4.24 ± 4.1 | **< 0.001** |
| Pain intensity (M ± SD) | 3.70 ± 1.2 | 4.76 ± 0.8 | **< 0.001** |
| Social support outside work (M ± SD) | 3.13 ± 0.8 | 3.58 ± 0.7 | **0.001** |
| Depression (M ± SD) | 5.46 ± 3.8 | 7.40 ± 5.0 | **0.01** |

*(Continued)*

**Table 1.** (Continued)

| Baseline characteristics | RTW (n = 94) | NRTW (n = 47) | *p*-value |
|---|---|---|---|
| Job strain (M ± SD) | 0.77 ± 0.2 | 0.86 ± 0.2 | **0.01** |
| Life-long pain duration (M, range), months | 81.63 (3–420) | 100.45 (4–360) | 0.32 |

[1]Examples of white-collar work include office administration, nursing and teaching; example of blue-collar work include elderly care, childcare, and cleaning.

[2]Coping through increasing behavioral activities and coping by ignoring sensations were measured using the Coping Strategies Questionnaire, with scores ranging from 0 to 31 points and higher values indicating more frequent use of the coping strategy.

[3]Beliefs about returning to the same work within 6 months were assessed using a single question and rated on a 10-point Likert scale (1 = highly unlikely to return to the same work; 10 = highly likely to return to the same work), M mean; SD standard deviation.

Table 2 shows the results of the multiple logistic regression analysis. All three predictors–coping through increasing behavioral activities, beliefs about returning to the same work within 6 months, and social support outside work–were significantly associated with RTW, and the results remained significant after controlling for age in the adjusted analysis. More specifically, women who more frequently used behavioral activities to cope with pain (OR: 1.14, 95% CI: 1.03–1.25) and more strongly believed they would return to the same work within 6 months (OR: 1.22, 95% CI: 1.10–1.37) had an increased probability of RTW at the 1-year follow-up. Women who had more social support outside work showed a decreased chance of RTW (OR: 0.50, 95% CI: 0.28–0.92). The regression model was statistically significant ($p < 0.001$), and 34% of the variance in the outcome variable was explained by the predictors.

## Discussion

The present results show that women who more frequently used behavioral activities as coping strategies and who more strongly believed they would return to the same work within 6 months had an increased chance of RTW, whereas women with higher social support outside work were less likely to RTW.

Coping through increasing behavioral activities such as leisure activities, reading and socialization was positively associated with RTW. This is consistent with a previous study suggesting that coping strategies such as relaxation, stress management, and activity training helped women on sick leave for MSP increase their ability to control and decrease the pain, which may be an important and effective tool for early RTW [59]. Another study found that coping with daily activities outside work was related to RTW among people with long-term MSP [20]. It may be that the women in the present study who had RTW were able to work despite the pain because they were coping with pain by increasing these behavioral activities. Indeed, the RTW group rated their pain intensity as 3.7 on average (scale of 0–6) and their behavioral activities as 13.1 on average (scale of 0–31), while the NRTW group rated their pain intensity as 4.8 and behavioral activities as 11.1 (Table 1). Consequently, further study is warranted to see whether pain intensity is a mediator in the relationship between behavioral activities and RTW in this population.

We found an association between beliefs about returning to the same work within 6 months and RTW. Our finding is consistent with an earlier study among women on sick leave showing that a positive expectation/belief regarding RTW within a year was associated with the ability to work and volition that facilitated RTW [18]. In addition, negative recovery beliefs have been found to be a risk factor for NRTW (i.e., sustained long-term sick leave) among individuals with non-specific chronic MSP [60]. Similarly, a systematic review investigating belief in recovery and its relationship with RTW [33] showed that positive work-related recovery beliefs

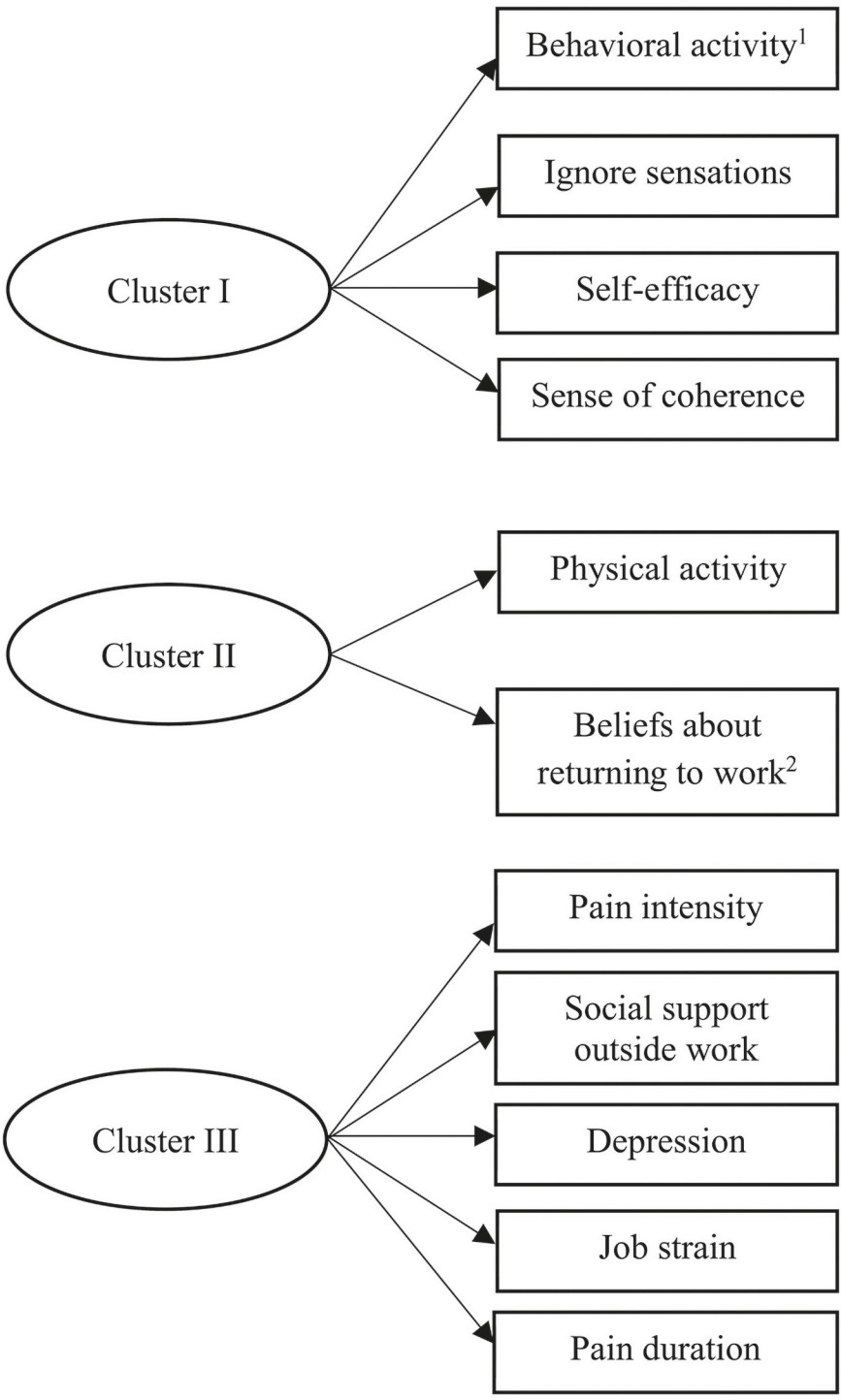

**Fig 2. Illustrates the three clusters.** [1]Coping through increasing behavioral activities; [2]Beliefs about returning to the same work within 6 months.

predicted RTW. Believing in RTW should perhaps be a goal in efforts to facilitate RTW among individuals on sick leave for long-term MSP [61, 62]. Such beliefs could be a target for change in rehabilitation focused on RTW for this group [19]. The reasoning here is that believing in RTW may change the individuals' attitudes, because attitudes originate from beliefs, which

**Table 2. Multiple logistic regression analysis of the selected predictors at baseline and return to work at 1-year follow-up.**

| Predictors | Unadjusted analysis | | | Adjusted analysis[1] | | |
|---|---|---|---|---|---|---|
| | SE | OR (95% CI) | *p*-value | SE | OR (95% CI) | *p*-value |
| Behavioral activity[2] | 0.05 | 1.12 (1.02–1.22) | 0.02 | 0.05 | 1.14 (1.03–1.25) | 0.008 |
| Beliefs about returning to work[3] | 0.06 | 1.24 (1.12–1.38) | < 0.001 | 0.06 | 1.22 (1.10–1.37) | < 0.001 |
| Social support outside work[4] | 0.31 | 0.49 (0.26–0.90) | 0.02 | 0.31 | 0.50 (0.28–0.92) | 0.03 |
| Age | | | | 0.03 | 0.94 (0.89–0.99) | 0.04 |
| *Overall model for the women* | $R^2 = 0.30$, $\chi^2 = 31.83$, $p < 0.001$ | | | $R^2 = 0.34$, $\chi^2 = 36.71$, $p < 0.001$ | | |

[1]Age was controlled for in the adjusted analysis;

[2]Coping through increasing behavioral activities, such as leisure activities, reading, and socialization was measured using the Coping Strategies Questionnaire, with scores ranging from 0 to 31 points and higher values indicating more frequent use of the coping strategy.

[3]Beliefs about returning to the same work within 6 months were assessed using a single question and rated on a 10-point Likert scale (1 = highly unlikely to return to the same work; 10 = highly likely to return to the same work).

[4]Social support outside work was measured using three items in the Multidimensional Pain Inventory, with scores ranging from 0 to 6 and higher values indicating higher social support.

SE standard error; OR Odds Ratio; CI Confidence Interval.

may increase their use of coping strategies such as behavioral activities that can help them manage their pain, thus supporting them in their RTW process.

Our results also indicate that social support outside work negatively predicted RTW. This result is in contrast to previous findings regarding social support and RTW. A qualitative study among workers on sick leave suggested that social support either from work or outside work was a positive indicator for RTW, although the authors recommended using quantitative measures of social support to verify their findings [30]. A quantitative study, however, found that social support from a partner might not be related to RTW among female workers on sick leave [32]. Regardless of this discrepancy, an explanation for our result could be that when individuals receive ample support from family and friends, they tend to feel comfortable about not returning to work. It is possible that having a great deal of social support may cause people to believe they no longer have the capacity to manage things without someone's help and this belief may exacerbate their pain-related fear. This may be the reason they avoid doing their daily activities [63], which could eventually impede their RTW. A reason for the contradiction between our results and previous findings could be that previous studies measured social support in general [30, 32], which covers all aspects in life as a whole, including the quality of the partner relationship, satisfaction, and social integration. However, in the present study, social support was assessed by pain-related social support items, covering pain-related support, concern, and consideration from family and friends [47]. In addition, social support from the workplace might be more relevant than social support from family in the RTW process.

The present findings are in agreement with results from previous studies on RTW among men and women with MSP, particularly for the predictors "coping through increasing behavioral activities" and "beliefs about returning to the same work within 6 months." However, this was not the case for the predictor "social support outside work," as our result was partly in agreement with previous findings among men and women with MSP. A quantitative study [31] investigating social support outside work (i.e., any type of support from family and friends) among men and women with musculoskeletal injury found that among women, receiving support from family was negatively associated with RTW whereas support from friends was positively associated with RTW; but no such association was observed among men [31]. Before giving a recommendation to investigate women separately in this regard, there is a

need for further research which uses larger samples and considers different measurements of social support in this population.

## Strengths and limitations

The strengths of the present study were the prospective design that included a 1-year follow-up of 68% of the participants, the selection of participants based on ICD-10 codes provided by a physician, and the fact that the predictors were selected without capitalizing on the relationship observed between predictors and the outcome.

One limitation of the study is its relatively small sample size, which allowed only a limited number of predictors to be considered simultaneously. Because the participants were not randomly selected, there may be sampling bias in the study, which may affect the external validity. A non-response analysis could not be conducted because we had no access to non-respondents' data, as the participants were invited by the Swedish Social Insurance Agency. It is unknown whether the participants had first returned to work and then relapsed to being on sick leave again before the follow-up measurement. It was also unknown whether the participants working < 50% at baseline had opportunities to receive support from the workplace. If such support was available, the participants who were working to some extent at baseline may have been more likely to RTW than the participants who did not work at all. Further, information was lacking on whether the participants received treatment, and whether they received or had been offered modified duties or support from supervisors/co-workers during the year between baseline and follow-up. Another limitation is that there may be common-method bias in the results because self-reported data were used to measure the predictors and outcome. Similarly, because data were measured subjectively, we were unable to ensure that the participants were only considering pain in the neck/shoulder and/or back when answering the questions. We had no information on the psychometric properties of the single-item tools used in the study (physical activity, belief about RTW within 6 months, and life-long pain duration). As the small sample size did not support more than two groups, the authors set the cut-off point of RTW at working > 50% of their service. This might have affected the external validity of the present study. Moreover, because all of the participants were women, the results cannot be generalized to men. Further studies should use larger samples to test expanded models of relevant predictors including potential mediating and/or moderating factors.

## Conclusions

The present study found that coping through increasing behavioral activities such as leisure activities, reading, and socialization as well as beliefs about returning to the same work within 6 months, increased the probability of RTW, whereas social support outside work decreased the chance of RTW at a 1-year follow-up among women with long-term neck/shoulder and/or back pain. The predictors highlighted here can be considered by healthcare professionals aiming to facilitate RTW in this population. Moreover, healthcare professionals should consider whether individuals are getting more social support outside work when supporting them in their RTW process.

## Supporting information

**S1 Fig. Illustrates dendrogram from cluster analysis.** [1]Behavioral activity = Coping through increasing behavioral activities; [2]Beliefs at work = Beliefs about returning to the same work within 6 months.
(PDF)

**S1 Table. Difference in mean at baseline data between participants and dropouts at 1-year follow-up by age, pain duration, pain intensity, work ability, and well-being.**
(PDF)

**S1 File.**
(PDF)

**S2 File.**
(PDF)

## Acknowledgments

We would like to extend our gratitude to all of the study participants, as well as to the Swedish Social Insurance Agency for assistance us in the data collection. We would also like to express our gratitude and appreciation to Marina Heiden (Associate Professor) for assisting us in developing the research design and statistical analysis.

## Author Contributions

**Conceptualization:** Mamunur Rashid, Marja-Leena Kristofferzon, Annika Nilsson.

**Data curation:** Mamunur Rashid, Marja-Leena Kristofferzon, Annika Nilsson.

**Formal analysis:** Mamunur Rashid, Marja-Leena Kristofferzon, Annika Nilsson.

**Funding acquisition:** Marja-Leena Kristofferzon.

**Investigation:** Marja-Leena Kristofferzon.

**Methodology:** Mamunur Rashid, Marja-Leena Kristofferzon, Annika Nilsson.

**Project administration:** Mamunur Rashid, Marja-Leena Kristofferzon, Annika Nilsson.

**Resources:** Annika Nilsson.

**Software:** Mamunur Rashid.

**Supervision:** Marja-Leena Kristofferzon, Annika Nilsson.

**Validation:** Mamunur Rashid, Marja-Leena Kristofferzon, Annika Nilsson.

**Visualization:** Mamunur Rashid, Marja-Leena Kristofferzon, Annika Nilsson.

**Writing – original draft:** Mamunur Rashid.

**Writing – review & editing:** Marja-Leena Kristofferzon, Annika Nilsson.

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
