## [Decision Letter · Decision Letter 0]

30 Jun 2021

PONE-D-21-16980

Predictors of return to work among women with long-term neck/shoulder and/or back pain: a 1-year prospective study

PLOS ONE

Dear Dr. Mamunur Rashid,

Thank you for submitting your manuscript to PLOS ONE. After careful consideration, we feel that it has merit but does not fully meet PLOS ONE’s publication criteria as it currently stands. Therefore, we invite you to submit a revised version of the manuscript that addresses the points raised during the review process.

Special attention should focus points 3, 4 and 5 raised by the referee due to their relevance. Accordingly, explain the difference between items chosen/listed by the pain severity MPI scale to infer pain intensity. Concerning the subscale selection from each questionnaire, why were pain catastrophizing (CSQ) and anxiety (HADS) not included since these are very relevant variables in pain experience? Concerning the rationale of the cluster analysis on all candidate predictors and then selection of one from each cluster, this must be better explained and justified. Specifically, the variables grouped under each cluster are in some cases very different: in cluster 3 pain intensity and social support do not assess similar constructs. In clinical terms this is highly speculative.

We look forward to receiving your revised manuscript.

Kind regards,

Armando Almeida

Academic Editor

PLOS ONE

Journal Requirements:

- https://bmcpublichealth.biomedcentral.com/articles/10.1186/s12889-021-10510-8

- https://bmcpublichealth.biomedcentral.com/articles/10.1186/s12889-018-5580-9

In your revision ensure you cite all your sources (including your own works), and quote or rephrase any duplicated text outside the methods section. Further consideration is dependent on these concerns being addressed.

Reviewers' comments:

Reviewer's Responses to Questions

**Comments to the Author**

1. Is the manuscript technically sound, and do the data support the conclusions?

Reviewer #1: Partly

2. Has the statistical analysis been performed appropriately and rigorously? 

Reviewer #1: I Don't Know

3. Have the authors made all data underlying the findings in their manuscript fully available?

Reviewer #1: No

4. Is the manuscript presented in an intelligible fashion and written in standard English?

Reviewer #1: Yes

5. Review Comments to the Author

Reviewer #1: 

This manuscript focuses on a relevant topic, given the high costs associated with pain-related sick leaves. However, I have some concerns that I have detailed below.

General comments:

1. Why did the authors opt to use the term "long-term pain" and not "chronic pain"? If the type of pain was labeled as chronic it would be clear that only patients with pain over 3 months were included (as per the IASP definition) and this could help clarify the sample in the study. For example, it is not clear in the abstract the pain duration considered for inclusion.

2. Globally, the paper is well written but would benefit from a revision for increased clarity in some sections.

Abstract:

1. I suggest adding information on where patients were recruited (social security agency).

2. In the Methods, it should be clear that clusters were used to identify one predictor from each "category".

INTRODUCTION

Page 4, line 121: "Increased work ability per se was found to be an important predictor for women with pain in the neck/shoulder and/or back" - It is not clear what work ability was a predictor of. Return to work? Or having pain?

Page 4, line 125: Please revise this sentence since it is a bit confusing.

METHODS

Major issues:

1. Did the authors control for insurance or litigation issues among the participants? In case of sick leave due to work-related injuries this is usually a relevant factor associated with RTW.

2. Is the cut-off for working or not working (50% from extent of employment) commonly used in these types of study or was it determined by the authors? Can some references be provided? It seems t me that, for example, it would not be appropriate to consider as RTW someone who was working 40% at baseline and 60% at follow-up.

3. The authors used the 3 original items from the pain severity MPI scale to infer pain intensity. However, this scale only has 2 items (suffering item was removed) according to the Swedish validation that the authors cite. What is the rationale to use 3 items? Providing a measure of reliability could provide a better sense of the adequacy of this scale, and I believe this data should be presented here, despite also being reported elsewhere.

Also, it s not clear which MPI subscale was used to assess "social support outside work". I suggest that the authors mention the name of the original subscales and provide reliability measures for all questionnaires used.

4. Concerning the subscale selection from each questionnaire, why were pain catastrophizing (CSQ) and anxiety (HADS) not included? These are very relevant variables in pain experience.

5. The authors perform a cluster analysis on all candidate predictors and then select one from each cluster, on the basis that they would provide similar information. However, I think that the rationale behind this decision must be better explained and justified. Though some variables may assess similar constructs, they are nonetheless different. In fact, the variables grouped under each cluster are in some cases very different: for example in cluster 3 pain intensity and social support do not assess similar constructs. Using this approach, relevant information may have been excluded from the analyses. For example, pain duration is very relevant to predict RTW. The problem of multicollinearity can be addressed by analyzing the VIF values of the models.

6. What criteria were followed to determine the number of predictors in the model? The authors state in the limitations that the sample size precluded the inclusion of more variables, but a sample size of 141 would allow for more than 3 DVs.

Minor issues:

Page 5, line 150: Is 65 years old the age of retirement in Sweden? Maybe this could be clarified in this section.

Page 5, line 156: In the cases where the myalgia code was used, was there any further specification to ensure that pain was located in the neck/shoulder/back?

Page 6, line 183: The reference for the CSQ is missing.

Page 6 and 7: The authors describe the CSQ and then state that they used two subscales from this questionnaire. However, when reporting the MPI the authors present its subscales separately. The same format of description should be adopted.

Page 7, 228-232: Is this a validated procedure? Can a reference be provided?

Page 7, line 234: Can the authors be sure that the participants answered this question only considering their neck/shoulder/back pain?

RESULTS

Major issues:

Page 9, line 273: Please describe the attrition analysis in the statistical analysis section. Also, these results should be supported by an accompanying table.

Page 9, line 279: The authors state that "the group who had RTW rated less pain intensity, shorter life-long pain duration and more strongly believed in returning to work within 6 months", but this is not supported by a formal statistical test.

Table 2. The logistic regression results should present Qui-squared statistics and R2.

Also, it would be important to have controlled for the effect of demographic variables such as age. I would expect that this variable has a strong influence on RTW.

DISCUSSION

Page 12, lines 322-324: Again, this comparisons must be based on a formal statistical test.

6. PLOS authors have the option to publish the peer review history of their article (what does this mean?). If published, this will include your full peer review and any attached files.

Reviewer #1: No

---

## [Author Response · Author response to Decision Letter 0]

30 Sep 2021

Response to the reviewer and editor comments has been attached in the attachment.

---

## [Decision Letter · Decision Letter 1]

7 Oct 2021

PONE-D-21-16980R1Predictors of return to work among women with long-term neck/shoulder and/or back pain: a 1-year prospective studyPLOS ONE

Dear Dr. Mamunur Rashid,

Thank you for submitting your manuscript to PLOS ONE. After careful consideration, we feel that it has merit but does not fully meet PLOS ONE’s publication criteria as it currently stands. Therefore, we invite you to submit a revised version of the manuscript that addresses the points raised during the review process.

Although the authors have answered all the concerns raised and responded adequately to most of them, the editor and referee consider that there are aspects of the work that deserve a better explanation. Please consider all points still raised by the referee. Importantly, (i) reasons to select specific subscales (CSQ and HADS) should be included in the text, (ii) as should how did the authors exclude patients with myalgia in body sites other than neck/shoulder/back; (iii) please include reference(s) that support the rationale for the cut-off set by the authors, and include this cut-off as a possible limitation in the corresponding section. (iv) the rationale for defining the clusters grouping very different variables should be clearly expressed in the text and studies validating the authors' approach should be included.

We look forward to receiving your revised manuscript.

Kind regards,

Armando Almeida

Academic Editor

PLOS ONE

Journal Requirements:

Additional Editor Comments (if provided):

Reviewers' comments:

Reviewer's Responses to Questions

**Comments to the Author**

1. If the authors have adequately addressed your comments raised in a previous round of review and you feel that this manuscript is now acceptable for publication, you may indicate that here to bypass the “Comments to the Author” section, enter your conflict of interest statement in the “Confidential to Editor” section, and submit your "Accept" recommendation.

Reviewer #1: (No Response)

2. Is the manuscript technically sound, and do the data support the conclusions?

Reviewer #1: Yes

3. Has the statistical analysis been performed appropriately and rigorously? 

Reviewer #1: I Don't Know

4. Have the authors made all data underlying the findings in their manuscript fully available?

Reviewer #1: Yes

5. Is the manuscript presented in an intelligible fashion and written in standard English?

Reviewer #1: Yes

6. Review Comments to the Author

Reviewer #1: I thank the authors for their careful replies.

I believe all issues were satisfactorily answered, but I still have some concerns that remain, related to the methodology of the study. The reasons to select specific subscales (CSQ and HADS) were explained, but I believe these should be clearly described in the text, so that each reader can judge its adequacy. Otherwise, it may seem somewhat arbitrary. For example, the reason given in the authors’ answer to not include pain catastrophizing (non-significant results) is not acceptable. These decisions should be theory-driven. The cut-off point for (non)return to work and the fact that participants could be performing part-time work at baseline remain an important limitation, but are mentioned in the corresponding section. I still feel that the clusters emerging from the analysis group very different variables. I suggest that the authors cite some studies using and validating this approach, to strengthen the robustness of this approach. Finally, I stress again that that manuscript would benefit from a thorough language revision.

ABSTRACT

I suggest: “Cluster analyses were performed to identify one variable from each cluster to enter the regression model”. Otherwise, it is stated that cluster analyses and multiple logistic regression were performed to identify predictors. Please revise the manuscript for accuracy in language.

INTRODUCTION

Page 4, line 120: “…individuals’ beliefs in their ability to work in the future and in their own influence in life as a whole are predictors of RTW”. The meaning of this sentence is not clear.

Page 4, line 125: “Social support…was positively associated with reduced MSP and stress”. This formulation is not clear. I believe that the authors mean that social support is associated with reduced MSP and stress. A positive association would imply that more social support was associated with more stress.

Page 4, line 126: “…could increase work ability and that could in turn give an importance for RTW”. Language is still not clear in this sentence. I give the following suggestion as an example to be considered by the authors “…could increase work ability which, in turn, could contribute to RTW”.

METHODS

Page 5, line 153: This sentence is confusing and its meaning is not clear.

Page 5, line 158: It is clear why the myalgia was code was used. But how did the authors exclude patients with myalgia in body sites other than neck/shoulder/back? In other words, how are the authors sure that only those patients complaining of myalgia in the neck/shoulder/back were included?

Page 6, line 188: As I understood it, all subscales were used but only these two were selected for the study. This should be clear.

Page 7, line 228: I believe it would be of interest to clarify in the text why the anxiety subscale was not used.

Page 7, lines 230 and 239: Notwithstanding considerations made on page 8, line 246, please ponder if it would be more consistent to also present alpha values in these sections, after describing each questionnaire.

Page 7, line 237: Afterwards was correct.

Page 8, line 251: It should be mentioned that this cut-off was set by the authors, and some studies using the same cut-off should be referenced if possible. The fact that this cut-off is a possible limitation should also be considered and mentioned in the corresponding section.

Page 9, line 284: I suggest adding a reference to S1 here to inform the reader about that table.

Page 10, line 296: To clarify this information, I suggest adding: “we chose three (plus one covariate) because…”

Page 10, line 298: Actually, the information added to is merely a repetition of what was already stated on page 9, lines 271-273. Therefore, this addition is not relevant to the content of the manuscript.

Table 2: Are the Adj R2, chi-square and p values for the adjusted or unadjusted models? Consider if both should be presented, for the reader to judge the proportion of variance explained by each model.

Limitations: The fact that possible litigation issues were not controlled should be mentioned as a possible limitation.

7. PLOS authors have the option to publish the peer review history of their article (what does this mean?). If published, this will include your full peer review and any attached files.

Reviewer #1: No

---

## [Author Response · Author response to Decision Letter 1]

2 Nov 2021

Respond to reviewers attached with files.

---

## [Decision Letter · Decision Letter 2]

11 Nov 2021

Predictors of return to work among women with long-term neck/shoulder and/or back pain: a 1-year prospective study

PONE-D-21-16980R2

Dear Dr. Rashid,

We’re pleased to inform you that your manuscript has been judged scientifically suitable for publication and will be formally accepted for publication once it meets all outstanding technical requirements.

Kind regards,

Armando Almeida

Academic Editor

PLOS ONE

Additional Editor Comments (optional):

Reviewers' comments:

Reviewer's Responses to Questions

**Comments to the Author**

1. If the authors have adequately addressed your comments raised in a previous round of review and you feel that this manuscript is now acceptable for publication, you may indicate that here to bypass the “Comments to the Author” section, enter your conflict of interest statement in the “Confidential to Editor” section, and submit your "Accept" recommendation.

Reviewer #1: All comments have been addressed

2. Is the manuscript technically sound, and do the data support the conclusions?

Reviewer #1: Yes

3. Has the statistical analysis been performed appropriately and rigorously? 

Reviewer #1: Yes

4. Have the authors made all data underlying the findings in their manuscript fully available?

Reviewer #1: Yes

5. Is the manuscript presented in an intelligible fashion and written in standard English?

Reviewer #1: Yes

6. Review Comments to the Author

Reviewer #1: (No Response)

7. PLOS authors have the option to publish the peer review history of their article (what does this mean?). If published, this will include your full peer review and any attached files.

Reviewer #1: No

---

## [Editor Report · Acceptance letter]

15 Nov 2021

PONE-D-21-16980R2 

Predictors of return to work among women with long-term neck/shoulder and/or back pain: a 1-year prospective study 

Dear Dr. Rashid:

I'm pleased to inform you that your manuscript has been deemed suitable for publication in PLOS ONE. Congratulations! Your manuscript is now with our production department. 

Kind regards, 

on behalf of

Prof. Armando Almeida 

Academic Editor

PLOS ONE